

# A modified parameterization of stratiform cloud microphysics for the Community Earth System Model

Chandra Shekhar Pant*[1], Deepak Waman[2], Sachin Patade[3], Akash Deshmukh[4], and Vaughan Phillips[3]

[1]Department of Hydro and Renewable Energy, Indian Institute of Technology Roorkee, Roorkee, India
[2]Institute of Meteorology and Climate Research, Department of Tropospheric Research, Karlsruhe Institute of Technology
[3]Department of Physical Geography and Ecosystem Science, Lund University, Lund, Sweden
[4]Atmospheric Research Centre of Eastern Finland, Finnish Meteorological Institute, Kuopio, Finland

**Abstract.** Large-scale stratiform clouds are widespread and dominate the Earth's radiation budget. Their radiative and microphysical properties are inseparable, depending on ambient aerosol conditions and on properties of any convective outflow. In the Community Atmospheric Model, version 6 (CAM6), large-scale clouds were originally treated two decades ago with a two-moment bulk microphysics approach. Since then, the technological and empirical basis of global models has improved, for example by representing cloud microphysics to encompass extra processes of ice and droplet initiation, including dependencies on aerosol conditions of size, composition, and loading.

To advance the microphysical realism of the large-scale cloud scheme of the global model CAM6, most of the known mechanisms of secondary ice production (SIP) and an empirical formulation for heterogeneous ice nucleation have been represented in the stratiform scheme of the Global model CAM6. We included a hybrid bin/bulk scheme that treats aerosol activation, growth processes of accretion, aggregation, and riming, and three SIP mechanisms in the stratiform cloud scheme. We simulated an observed case of a mesoscale convective system during the Mid-latitude Continental Convective Clouds Experiment (MC3E) in Oklahoma, USA, using the Single-Column Atmosphere Model (SCAM6). The results from the simulations are validated against the aircraft, satellite, and ground measurements.

Results show that the modified stratiform scheme can predict the cloud properties of the observed stratiform clouds realistically. Together with our improved convective scheme in CAM6, this paves the way for more realism in the treatment of aerosol cloud interaction in global climate change by conventional General Circulation Models.

## 1 Introduction

Cloud parameterisations are essential for climate and weather prediction in conventional General Circulation Models (GCMs). GCMs partition the problem of cloud parameterisation into two distinct broad categories, namely for stratiform and convective clouds. Convective clouds have a strong updraft velocity ($> 1$ m/s), and stratiform regions have weak ascent ($< 1$ m/s) with less spatial variability (Houze, 1989; Sui et al., 2007). Convective precipitation is characterised by short duration, high intensity, and rapid fluctuations, on similar scales to the related convective clouds, which are unresolved by conventional global models.

---

*Correspondence to: Chandra Shekhar Pant (email: csp@hre.iitr.ac.in)



Stratiform precipitation is more long-lasting and widespread than convective rain, resulting in significant rainfall accumulation (Houze, 2014).

Stratiform clouds are extensive and characterized by minimal vertical motion, often covering regions of up to 1000 km in horizontal distance. Their large-scale average properties can be treated as prognostic variables and resolved by global models, although most of the variability of their properties remains unresolved. These clouds greatly influence the Earth's radiative balance by reflecting sunlight back into space and interacting with longwave radiation, resulting in a net cooling effect (Liou, K. N., 2002 "An Introduction to Atmospheric Radiation" Academic Press). Simulations from cloud-resolving models accurately estimate cloud properties but tend to underestimate stratiform precipitation (Fridlind et al., 2017; Varble et al., 2014). Various studies have shown that under-prediction in stratiform precipitation by cloud models may be due to biases in treatment of the raindrop size distribution (Li et al., 2009), underestimation of ice water content (Varble et al., 2014), or lack of detrained convective outflow (Bryan and Morrison, 2012).

Microphysical processes involve the conversion of water vapour to different types of hydrometeors in clouds and the transfer of mass among these different types. Liquid hydrometeors are cloud droplets and raindrops; ice hydrometeors are snow, graupel/hail, and cloud ice. Aerosols in the atmosphere act as cloud condensation nuclei (CCN) and a tiny minority of them act as ice nucleating particles (INPs) to initiate cloud droplets and ice crystals, respectively (Petters and Kreidenweis, 2007; Andreae and Rosenfeld, 2008). Secondary ice production (SIP) enhances ice number concentrations from pre-existing ice precipitation particles independently of any aerosol influence (Field et al., 2016; Yang et al., 2016; Korolev and Leisner, 2020). Since the ice and liquid phases in mixed-phase clouds are inter-related through the Bergeron-Findeisen process and by coagulation processes of growth in nature, it is essential to include all ice initiation mechanisms to accurately predict cloud phase and radiative properties, which subsequently influence cloud coverage and longevity (Sun and Shine, 1994; Field and Heymsfield, 2015; Mülmenstädt et al., 2015; Gupta et al., 2023).

Many observations have shown that ice number concentrations are typically up to four orders of magnitude higher than active INP concentrations for cloud-top temperatures (Harris-Hobbs and Cooper, 1987; Ladino et al., 2017; Lasher-Trapp et al., 2021). Recent studies, including various SIP mechanisms, have demonstrated an improvement in the prediction of ice number concentrations (Sullivan et al., 2017, 2018; Sotiropoulou et al., 2020, 2021; Phillips et al., 2017a; Waman et al., 2022a). SIP mechanisms influence the cloud properties such as cloud lifetime, precipitation rate and electrification (Phillips et al., 2017b, a, 2020; Sotiropoulou et al., 2021). Some of the proposed SIP mechanisms (Field et al., 2016) are

1. The Hallett-Mossop ("HM") process of rime splintering (Hallett and Mossop, 1974)

2. Fragmentation during ice-ice collisions (Vardiman, 1978; Takahashi et al., 1995; Yano and Phillips, 2011; Gautam et al., 2024; Jadav et al., 2025)

3. Fragmentation during raindrop freezing (Dye and P. V. Hobbs (1968))

The HM process is currently controversial as a recent lab study failed to observe it (Seidel et al., 2024). The reason might conceivably be that artificial subsaturation with respect to ice in the airflow around the rimer might have depleted HM splinters before they could be detected in this recent experiment.





The paper shows an improvement in the treatment of the interaction between large-scale cloud properties and the aerosol conditions of the environment. A new treatment of mechanisms for cloud droplet and ice crystal initiation is included in the framework of the existing stratiform cloud microphysics scheme by Morrison and Grabowski (2008). We include all of the SIP mechanisms noted above, except for the HM process, which is already treated. Section 2 describes the model and the new microphysical processes represented in it. Section 4 presents the single-column model results using the new scheme, compared with coincident observations and the original stratiform scheme. The main conclusions of this study are summarised in the concluding section (Sec. 6).

## 2 Model description

### 2.1 Community Earth System Model (CESM)

The Community Atmospheric Model, version 6 (CAM6), is the atmospheric component of CESM. The Single Column Atmospheric Model, version 6 (SCAM6) represents a single grid-box column of the global model and utilises the CAM6 physics package (Gettelman et al., 2019). Large-scale tendencies for SCAM6 are prescribed from observations or global simulations. SCAM6 is a valuable tool for developing and testing parameterisations generally.

In CAM6, the original version of the stratiform cloud microphysics scheme followed a two-moment bulk microphysics approach (Morrison and Grabowski, 2008), hereafter 'MG08'. The scheme represented four cloud hydrometeor species: cloud liquid, cloud ice, snow and rain. The activation of cloud droplets followed Abdul-Razzak and Ghan (2000). Initiation of ice crystals followed Liu and Penner (2005). The ice number concentration in the stratiform microphysics scheme was limited so as not to exceed the so called "prescribed" value calculated at $-35°C$. For SIP, the original stratiform scheme represented only the HM process of rime-splintering, and other SIP processes were omitted Liu and Penner (2005).

### 2.2 Modified representation of cloud microphysics

In this paper, we have modified the stratiform cloud microphysics in the CAM6 global model by including new microphysical processes and changing the representation of certain included processes of stratiform microphysics. The changes are summarised as follows:

1. The microphysics scheme now represents five cloud hydrometeor species: cloud droplets, rain, cloud-ice, snow, and graupel/hail. Graupel/hail mass and number mixing ratios are treated diagnostically instead of prognostically for the purpose of treating microphysical processes such as SIP (HM process, breakup in ice-ice collisions). Since there is no prognostic variable for graupel/hail in the global model (CAM6), its amount is diagnosed for the purpose of treating microphysical processes, according to a look-up table for graupel mass (as a function of cloud-liquid and snow mass mixing ratios and temperature) from high-resolution cloud simulations with AC.

2. The cloud base droplet activation is represented by a scheme following Ming et al. (2006), which is more accurate for the treatment of aerosol conditions of chemistry.





3. In-cloud droplet activation of aerosol species is now represented by $\kappa$-Kohler theory because it allows internal mixtures
   (e.g. dust or BC coated with soluble aerosols) to be treated accurately (Petters and Kreidenweis, 2007).

4. Three extra SIP mechanisms noted above (sect. 1) are included.

5. Growth processes of aggregation, accretion and riming are treated with emulated bin microphysics schemes.

The treatment of stratiform microphysics is qualitatively consistent with the new convective microphysics scheme (Jadav et al. 2025). Both schemes being related to the microphysical treatment in a high-resolution aerosol-cloud (AC) model (Phillips et al., 2007, 2009, 2018; Kudzotsa et al., 2016).

### 2.2.1 Treatment of aerosol

The aerosol treatment follows Phillips et al. (2009) consisting of seven chemical species of aerosols, including both soluble and solid species of aerosol material. The soluble aerosol species consist of larger and smaller modes of sulphate (both modes are resolved explicitly), sea salt, and secondary organic matter. The solid aerosol species consist of mineral dust, black carbon, non-biological primary organics, and biological primary organics. Aerosols are depleted by wet scavenging and in-cloud nucleation.

Log-normal size distributions are implemented for all aerosol chemical species (Pruppacher and Klett, 2010). The distribution parameters of the aerosols are given in Table 1

**Table 1.** Aerosol properties. A comma separates the modes

| Aerosol group | Number of modes | $\log_{10}$ of standard deviation | Geometric mean diameter ($\mu$m) |
|---|---|---|---|
| Sulphate | 2 | 0.20, 0.30 | 0.8, 3 below 2km in the planetary boundary layer (PBL). Height-dependent formula is used above PBL. |
| Seasalt | 3 | 0.04, 0.16, 0.25 | 0.03, 0.18, 4.4 |
| Soluble organics | 2 | 0.27, 0.30 | 0.05, 0.04 below PBL and a height dependent formulae above. |
| Mineral dust | 2 | 0.27,0.20 | 0.8,0.3 |
| Black carbon | 1 | 0.20 | 0.09 |
| insoluble | 1 | 0.20 | 0.2 |
| biological | 2 | 0.40,0.59 | 0.16,0.46 |





### 2.2.2 Particle size distribution of hydrometeors

Representations of new microphysics processes follow a hybrid bin/bulk approach.

### 2.2.3 Bulk approach

The bulk parameterisation follows a gamma size distribution (Phillips et al., 2007, 2017b)

$$n_x = \int_0^\infty n(D_x) dD_x / \rho \tag{1}$$

The mass mixing ratio for cloud liquid, cloud-ice is given as,

$$q_x = \frac{\pi}{6} \int_0^\infty \rho_x D_x^3 n(D_x) \, dD_x / \rho \tag{2}$$

Here $x = w, i, r, s, g$ represent hydrometeor species. $n_x$ and $q_x$ are the hydrometer's number and mass mixing ratios. $\rho_x$ is the bulk density of the hydrometer. $D_x$ is the equivalent spherical diameter and $n(D_x) dD_x$ is the number concentration ($m^{-3}$) of cloud hydrometeor in size range $dD_x$.

$$\lambda_x = \left[ \frac{\Gamma(4 + p_x) \rho_x \frac{\pi}{6} n_x}{\Gamma(1 + p_x) q_x} \right] \tag{3}$$

Here, $\Gamma$ is the gamma function. For snow and graupel/hail, the bulk density is a function of size.

### 2.2.4 Emulating bin approach

The emulating bin approach is implemented for riming, accretion and aggregation growth processes.

33 temporary size bins are created to discretise particle size distributions. The mass in the smallest bin is calculated according to the smallest diameter.

$$m_{x,1} = \frac{\pi \rho_x D_{x,1}^3}{6} \tag{4}$$

The subscript $x = w, i, s, r, g$ represents cloud liquid, cloud-ice, snow, rain and graupel/hail, respectively. $\rho_x$ is the bulk density of the hydrometeor. $D_x$ is the spherical equivalent diameter. The mass in each size bin is equal to the previous bin multiplied by a specific factor. The temporary grid of size and mass bins is fixed.

$$n(D_x) = n_{x,o} D_x^{p_x} e^{-\lambda_x D_x} \tag{5}$$

$p_x$ represents the shape parameter. For cloud liquid $p_w = 3.5$, cloud-ice $p_i = 1$, rain $p_r = 2.5$, graupel/hail $p_g = 1$ (Phillips et al., 2007). The shape parameter for snow is calculated from a lookup table that takes into account the size dependence of





bulk density and axial ratio (Heymsfield et al., 2002). $\lambda_x$ represents the slope of the size distribution. More details are given by Phillips et al. (2017a) and Phillips et al. (2020). The number mixing ratio is not prescribed and its increment is predicted for each process of ice initiation.

### 2.2.5 Cloud droplet activation

The representation of cloud droplets activated at cloud base follows Ming et al. (2006). This scheme links the droplet number concentration to the aerosol size and chemistry. In timestep, $\Delta t$, the increments of droplet number and mass mixing ratios are:

$$\Delta n_c = -\sum_{i=1}^{4}\sum_{j=1}^{j=20} \Delta N_{aerosol}(i,j)$$

$$\Delta q_c = -\sum_{i=1}^{i=4}\sum_{j=1}^{j=20} \frac{\pi}{6} \Delta N_{aerosol}(i,j) D_{pmax}{}^3(i,j)\rho_w$$

(6)

$i$ labels the aerosol species from the soluble aerosol group (sulphate in both modes, secondary organic matter, sea salt), and $j$ is for the size bins of $i^{th}$ aerosol species. $\Delta q_c$ and $\Delta n_c$ are the mass and number of activated cloud droplets. $\Delta N_{aerosol}(i,j)$

is the number of activated aerosol and $D_{pmax}$ is the droplet diameter at maximum supersaturation in the $j^{th}$ bin for $i^{th}$ aerosol species.

Cloud-base activation of droplets occurs at the lowest level in-cloud (defined by thresholds on cloud-droplet concentration and cloud-liquid mass). At all other in-cloud levels, in-cloud droplet activation follows the $\kappa$-Kohler theory (Petters and Kreidenweis, 2007), and the supersaturation is approximated with the time-dependent analytical formula including dependencies

on the total mass and number concentrations of cloud-droplets and ice particles ((Korolev and Mazin, 2003) Equations 11-13). In-cloud droplet activation of soluble aerosols and insoluble aerosols coated with soluble material is treated for the various aerosol species of CAM, with the above emulated bin system for each species. The global model time-step is split up into many sub-cycles, each typically of duration of 10% that of the relaxation time-scale, and the in-cloud activation is performed on each sub-cycle. Over the sub-cycles, the droplet number is predicted to evolve from the activation while the cloud-liquid

mass is assumed constant, while the evolution of the cloud-droplet mean size is diagnosed. In each sub-cycle the evolving supersaturation is predicted.

### 2.2.6 Heterogeneous ice nucleation

The Empirical Parameterisation (EP) developed by Phillips et al. (2008, 2013) has been implemented in the scheme. EP is based on coincident field observations of the INP activity and the loadings of insoluble aerosol particles in the troposphere

from the Ice Nuclei Spectroscopy (INSPECT) campaign (DeMott et al., 2003; Richardson et al., 2007). The classification of concentrations of ice nuclei among aerosol types (dust, metallic compounds, inorganic black carbon, and insoluble organic aerosols) is informed by observations. Thus, the parameterisation can reflect the diversity of aerosol chemistry in the environment. The EP includes modes of immersion freezing, deposition freezing and condensation followed by freezing, which are treated here.





The number mixing ratio of cloud-ice particles, $\Delta n_i$, generated each time-step is given by

$$\Delta n_i = \sum_{X'} \max(n_{IN,X'} - n_{X',a}, 0) \equiv \sum_{X'} \Delta n_{X',a} \tag{7}$$

Here, $X$ represents the solid aerosol group consisting of dust, black carbon, insoluble non-biological organic matter, and primary biological organic matter. $n_{IN,X'}$ is the number of INPs activated by deposition and condensation/immersion-freezing modes from the group $X'$. $n_{X',a}$ is the number mixing ratio of INPs lost by activation as ice particles from group $X'$.

The supersaturation with respect to ice is an input to the EP scheme and is estimated as follows. If the cloud is liquid-only without ice, then water saturation is assumed. If the cloud contains ice, then an analytical expression for the time evolution of the supersaturation during the time-step is obtained from the equilibrium supersaturation with dependencies on concentrations and mean sizes of cloud-droplets and ice particles (Korolev and Mazin 2003, their Equations 11-13), though without sub-cycling.

### 2.2.7   Homogeneous freezing

Homogeneous freezing of supercooled cloud droplets and rain at $-35°C$ is treated following Phillips et al. (2007, 2009). Homogeneous freezing of solute aerosols is included with dependencies on humidity, temperature, and aerosol dry size for each aerosol species. There is sub-cycling throughout the global model time-step (50 sub-cycles per global model timestep), with the Lagrangian tracing vertically in 1D of an adiabatic parcel initiated at each level. The parcel ascends at constant ascent and the temperature change for each subcycle of ascent is prescribed from the ice-saturated adiabatic lapse rate. During each sub-cycle, the pre-existing ice and newly nucleated ice are each treated with temporary bulk variables of mass and number of particles, with their vapour growth explicitly treated (Rogers and Yau, 1996). The humidity is predicted explicitly inside the parcel, informing the homogeneous aerosol freezing routine every sub-cycle. At the end of the global model time-step, the total mass and number of newly nucleated ice particles are then transferred to the global model grid and aerosol amounts are depleted accordingly.

### 2.2.8   Hallett-Mossop or Rime Splintering

The representation of the HM process follows Phillips et al. (2007) and Kudzotsa et al. (2016). The HM process is active between $-3°$and $-8°C$. 350 ice splinters are produced for 1 mg of supercooled cloud liquid accreted onto snow or graupel/hail. The observed dependency on mean diameter of the cloud-droplets is accounted for.

### 2.2.9   Fragmentation during ice-ice collisions

Breakup in ice-ice collisions is treated following Phillips et al. (2017b) based on the principle of energy conservation Phillips et al. (2017a). Fragments smaller than 0.3 mm are added to the cloud-ice category; otherwise, they are added to snow. Size distributions of colliding ice particles are discretised in size bins, with their concentration represented in each bin using the emulated bin approach (sect. 2.2.4). The breakup scheme is applied to collisions in all permutations of pairs of bins of the interacting ice crystals. Three types of collisions are considered:





1. graupel/hail with other graupel/hail.

2. snow with other snow or graupel/hail.

3. graupel/hail with cloud-ice.

For two colliding particles, the change in the mass and number mixing ratio of the particles receiving the fragments is calculated
as,

$$\Delta n_i \approx \mathcal{N} \delta n_1 \delta n_2 \pi (r_1 + r_2)^2 |v_1 - v_2| E_c \Delta t / \rho$$
$$\Delta Q_i = \Delta n_i \zeta m_l \tag{8}$$

where $\mathcal{N}$ is the number of fragments per collision. Here $\delta n_1$ and $\delta n_2$ are the number concentrations of colliding particles in
size bins $\delta r_1$ and $\delta r_2$ with fall speeds, $v_1$ and $v_2$, and masses, $m_1$ and $m_2$. Also $E_c$ is the collision efficiency. Also, $l = 1\,or\,2$,
depending on which particle is more fragile. For collisions among graupel/hail particles, the particle with a smaller maximum
diameter is considered to be the more fragile one. For other types of collisions (points 2 and 3 above), cloud-ice and snow are
assumed to be more fragile. For the more fragile particle of the colliding pair, $\zeta$ is the ratio of the initial mass per fragment to
its parent mass. $\Delta n_i$ and $\Delta Q_i$ are summed over all permutations of size bins of colliding particles. $\sum \Delta n_i$ and $\sum \Delta Q_i$ are
the total increments of number and mass mixing ratio. $\sum \Delta Q_i$, the total mass mixing ratio of fragments is deducted from the
fragile colliding species. More details are provided in Phillips et al. (2017a).

### 2.2.10 Fragmentation during raindrop freezing

An empirical formulation for raindrop freezing fragmentation in two modes from Phillips et al. (2018) is applied,

1. Mode 1: Supercooled raindrop $(0.05 - 5\text{ mm}$ in diameter) collides with a smaller crystal or freezes heterogeneously.

2. Mode 2: Supercooled raindrop collides with more massive ice crystal emitting splashes, which produces secondary ice.

Supercooled drops are discretised in size bins according to Sec.2.2.4 with their concentration represented in each size bin.
For a supercooled drop colliding with ice crystal (cloud-ice, snow, graupel/hail), the number mixing ratio from drop freezing,
$\Delta(\delta n)$ in time step $\Delta t$ is predicted as:

$$\Delta(\delta n) \approx -E_c(D, D_i) \delta \tilde{n} \delta \tilde{n}_i \pi \left( \frac{D}{2} + \frac{D_i}{2} \right)^2 |v - v_i| \Delta t / \rho \tag{9}$$

The subscript $i$ denotes the ice hydrometer receiving the fragments. $E_c$ is the collision efficiency, and $\rho$ is air density. $v$ and $v_i$
are the fall velocities of supercooled drops and ice crystals. $\tilde{n} = \rho n$ and $\tilde{n}_i = \rho n_i$, where tilde denotes the number concentration
per unit volume.

The change in number and mass bulk mixing ratios from drop freezing in size bin, $\delta D$, is given as,

$$\Delta n_i = -N \Delta(\delta n) \tag{10}$$



**Table 2.** The microphysical conversion tendencies for mass mixing ratio ($\mathrm{kg^{-1}kg^{-1}s^{-1}}$). The first symbols within the parentheses before the semicolon represent the final species in each interaction. The symbols after the semicolon represent the interacting species. The table is a modified version of Phillips et al. (2007, Table 1.).

| Symbol | Meaning |
|---|---|
| Ac $(q_g, q_i; q_c\|q_i)$ | Riming of cloud droplet by cloud-ice. |
| Ac $(q_g, q_i; q_g\|q_c)$ | Riming of cloud droplet by graupel/hail. |
| Ac $(q_s, q_i; q_s\|q_c)$ | Riming of cloud droplet by snow. |
| Ac $(q_c, q_r, q_i, q_g; q_r\|q_g$ ) | Accretion of rain by graupel/hail. |
| Ac $(q_s; q_i\|q_s)$ | Accretion of cloud-ice by snow |
| Ac $(q_c, q_s, q_g, q_i; q_s\|q_r)$ | Accretion of snow by rain. |
| Ac $(q_g, q_i; q_i\|q_r)$ | Accretion of cloud-ice by rain. |
| Ag $(q_i; q_i\|q_i)$ | Aggregation of cloud-ice and cloud-ice. |
| Ag $(q_s; q_s\|q_s)$ | Aggregation of snow and snow. |
| Ag $(q_g; q_g\|q_g)$ | Aggregation of graupel/hail and graupel/hail. |
| Ag $(q_g, q_s; q_g\|q_S)$ | Aggregation of graupel/hail and snow. |
| Ag $(q_g; q_g\|q_i)$ | Aggregation of graupel/hail and cloud-ice. |

$$\Delta q_i = -(N_B m_B + N_T m_T)\Delta(\delta n) \tag{11}$$

$i$ denotes the ice hydrometeor receiving the fragments. Fragments smaller than 300 $\mu$m are added to cloud ice; otherwise, they are added to snow or graupel/hail. $N_T$ are big secondary fragments per frozen drop, and $N_B$ are tiny secondary fragments per frozen drop. $m_T$ and $m_B$ are the initial mass of tiny and big ice fragments.

### 2.2.11 Growth Processes

The growth processes in Table 2 are now included in the model.

The emulated bin approach given in Sec.2.2.4 is applied. For two interacting particles, $x$ collecting $y$, the change in the mass mixing ratio per unit time is:

$$\frac{\Delta q_{x,y}}{\Delta t} = \sum_{i=1}^{i=33}\sum_{j=1}^{j=33} \chi_{x,y}(i,j)n_x(i)n_y(j)m_y(j) \tag{12}$$

The change in the number mixing ratio is,

$$\frac{\Delta n_{x,y}}{\Delta t} = \sum_{i=1}^{i=33}\sum_{j=1}^{j=33} \chi_{x,y}(i,j)n_x(i)n_y(j) \tag{13}$$



$i$ and $j$ are the indices for discretised size bins. $\chi_{x,y}$ is the collection kernel. $n_x(i)$ is the number mixing ratio of $x$ in $i^{th}$ size bin. $n_y(j)$ and $m_y(j)$ are the number and mass mixing ratio of $y$ in $j^{th}$ size bin. More details for the calculations of the collection kernels are provided in Phillips et al. (2005, 2015).

Turbulent-induced enhancement of accretion is treated in the newly included processes using the approach by Benmoshe and Khain (2014). More details are provided in Kudzotsa et al. (2016).

## 3 Methodology for model validation with an observed case

### 3.1 Mid-latitude Continental Convective Clouds Experiment (MC3E) campaign

The MC3E campaign was carried out over the Atmospheric Radiation Measurement (ARM) Southern Great Plain (SGP) in Oklahoma to study mesoscale convective system (MCS) from April to June 2011. The campaign consisted of a Central facility (CF) and 20 extended facilities, which covered an area with a 150 km radius. The campaign incorporated ground-based and in situ aircraft observations (Jensen et al., 2016).

### 3.1.1 Overview of observed storm on 11 May 2011

An MCS was initiated by a surface cold front with a parallel stratiform region north of a convective line (Jensen et al., 2016). The MCS storm consisting of this line of convective clouds was observed during 0900 to 2400 UTC on 11 May 2011. The storm had transitioned to a convective line with trailing stratiform cloud as it passed over the CF. The microphysical properties observed by aircraft were similar to those seen for trailing stratiform regions generally (Jensen et al., 2016).

### 3.1.2 Aircraft observations

Flights by the National Aeronautics and Space Administration (NASA) ER-2 and the University of North Dakota (UND) Cessna Citation 2 aircraft sampled the MCS between $18:00$ and $21:00$ UTC. The Citation carried probes to measure cloud microphysical properties.

CDP measured the sizes and number concentration of cloud-droplets, and their liquid water content (LWC). 2DC, CIP and HVPS$-3$ measured the ice concentrations. The combined ('COMB') spectrum includes the particle size distributions from 2DC (or CIP) and HVPS-3 probes. Shattering corrected tips were present for 2DC and HVPS$-3$ probes (Korolev et al., 2011) but not for the CIP. Following the method by Field et al. (2006) and Korolev et al. (2011), only ice crystals greater than 200 $\mu$m are considered for both observations and simulations in the validation plots.

### 3.1.3 Ground-based measurements

Xie et al. (2014) used constrained variational analysis to derive large-scale advective tendencies of heat and moisture and corresponding surface fluxes. These were applied to drive the simulations. Concentrations of active cloud condensation nuclei (CCN) were measured at seven supersaturation levels (Uin, 2016).





**Table 3.** Instruments used to measure cloud properties carried on Citation 2 aircraft

| Instruments | Measurement range |
|---|---|
| Cloud Droplet Probe (CDP) | $2 - 50 \ \mu m$ |
| King hot-wire Liquid water content probe | $0.01 - 5 \ \mathrm{g \ m^{-3}}$ |
| Nevzorov probe | $0.03 - 3 \ \mathrm{g \ m^{-3}}$ |
| Cloud Imaging Probe (CIP) | $0.025 - 1.5 \ \mathrm{mm}$ |
| 2D Cloud Imaging Probe (2DC) | $0.03 - 1.0 \ \mathrm{mm}$ |
| High-volume precipitation spectrometer, version 3 (HVPS-3) | $0.15 - 19.2 \ \mathrm{mm}$ |

## 3.2 Model Setup

The observed case of the MCS was simulated from 10 May 2011 to 13 May 2011 in SCAM6 with a global time step of 20 minutes and grid size of $100 \times 100$ km. Aerosol concentrations in each species are initially determined from the Goddard Chemistry Aerosol Radiation and Transport (GOCART) global model (Chin et al., 2000). These vertical profiles of aerosol concentration were then adjusted at all levels based on the averaged measurements near the ground from the Interagency Monitoring of Protected Visual Environments (IMPROVE) from 9 to 12 May 2011. More details of the initial aerosol conditions

are provided in (Waman et al., 2022b).

## 4 Results from the control simulation with a new stratiform scheme

### 4.1 Validation of new scheme with observations

Simulations are performed with the new stratiform scheme referred to as 'LS25'. The predicted cloud hydrometer profiles are conditionally averaged over the simulation period and are validated against aircraft data. Regarding the aircraft observations,

the stratiform region is considered to be where vertical velocity is less than 3 m s$^{-1}$. We also compare the results from the model simulations with the original MG08 scheme.

Figure 1a shows that the LS25 run predicts the two precipitation peaks at 2100 UTC 10 May and 18 UTC 11 May with realistic timing but insufficient intensity by a factor of 2 (weaker initial peak) and by about 20% (main peak) respectively. A consequence is a surplus of humidity in the environment remains to allow too much stratiform cloud after the main peak on 12

May. Regarding the cumulative surface precipitation, this is over-predicted by about 10% at the end of the simulation for LS24. Stratiform precipitation overall contributes 65% to the predicted accumulated precipitation with the LS24 run, in agreement with detailed simulations of the case by AC (Gupta et al. 2023).

Table 4 shows that the LS24 run has an error of about 20% to 30% in the radiative fluxes at TOA and at the surface when compared to the satellite observations. An under-prediction by 17% of net shortwave radiative flux at TOA entering the climate



system is consistent with the model over-estimating the amount of cloud condensate. Insufficient outgoing longwave radiation at TOA by $20\%$ is explicable in terms of cold cloud that is too high. It can be inferred from Figure 3, a significant amount of ice crystals exists at upper levels because of homogeneous freezing. Higher-level clouds with abundant ice crystals act to reduce the longwave radiation emitted to space. However, the radiation predictions are more accurate than for the MG08 scheme.



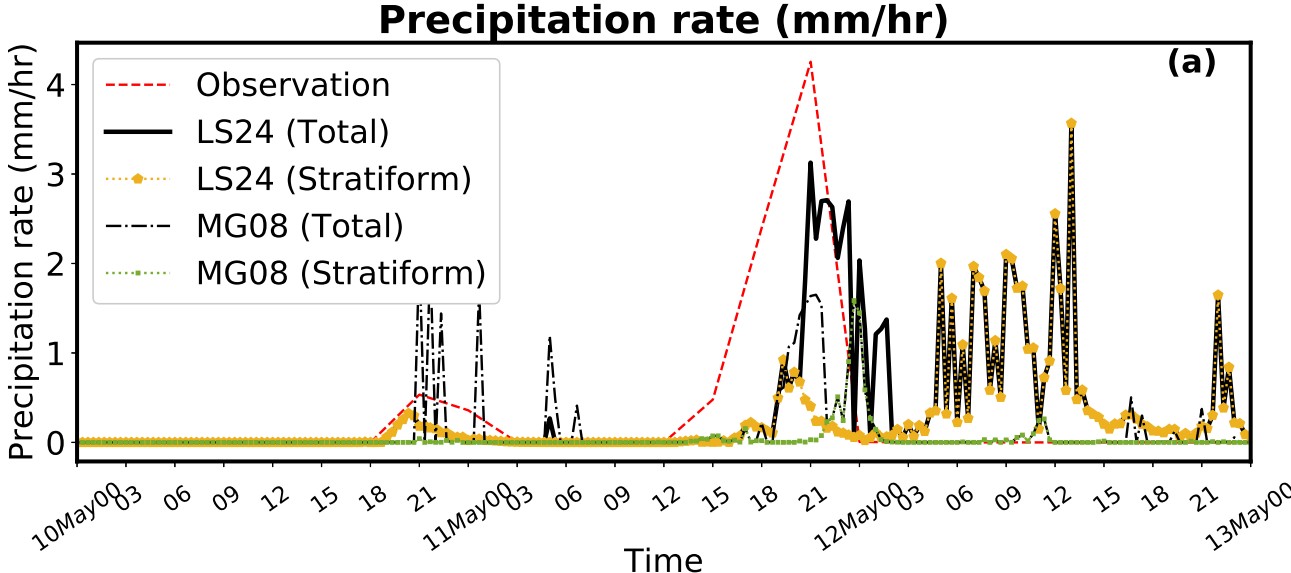

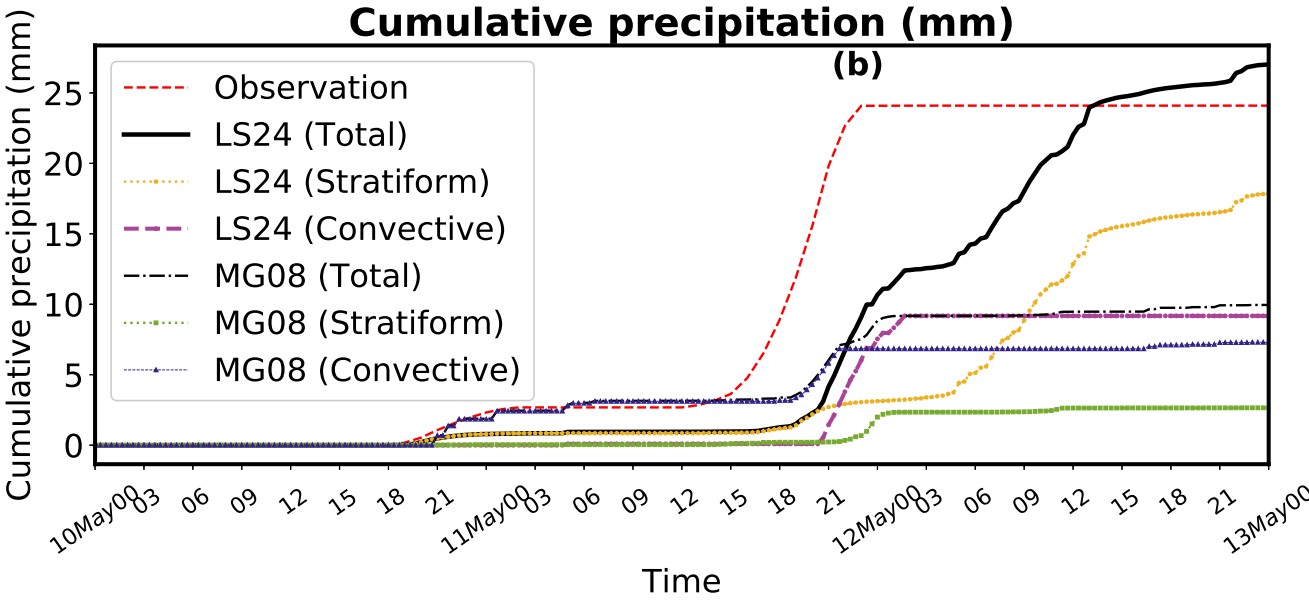

**Figure 1.** Comparison with MC3E observations of the domain-wide average predictions, from the MG08 (dash-dotted black line) and LS24 (solid black line) simulations, of (a) precipitation rate (mm/hr) and (b) cumulative surface precipitation (Xie et al., 2014). Also shown are the components of these predictions from stratiform precipitation (faint dotted lines).





**Table 4.** Unconditional average of the radiative fluxes for the simulation period from 00 UTC 10 May 2011 to 00 UTC 13 May 2011.

| Radiation fluxes (W/m$^2$) | Net Shortwave (SW) radiative flux at TOA | Net Longwave (LW) radiative flux at TOA | Net Shortwave radiative flux (SW) at surface | Net Longwave (LW) radiative flux at surface |
|---|---|---|---|---|
| Observations | 320.5 | 247.8 | 207.36 | 67.5 |
| LS24 | 266.68 | 196.55 | 149.44 | 49.11 |
| MG08 | 230.82 | 214.07 | 122.49 | 51.62 |

Figure 2a shows that the LS25 run predicts the cloud base is at $16°$C, in agreement with detailed simulations by AC ((Waman et al., 2022b)). The LWC predicted by the LS24 run, agrees well with the aircraft observations and falls within the $90\%$ confidence intervals for mean values. The LS24 run predicts a maximum LWC of $0.5$ gm$^{-3}$ at $14°$C. The observed LWC data-points differ less from the LS24 simulation than they do from each other.

Figure 2b shows the cloud droplet number concentration (CDNC) in comparison with the probe data (King, Nevzorov and CDP probes). CDNC predicted by the LS24 run agrees with aircraft observations at most levels.




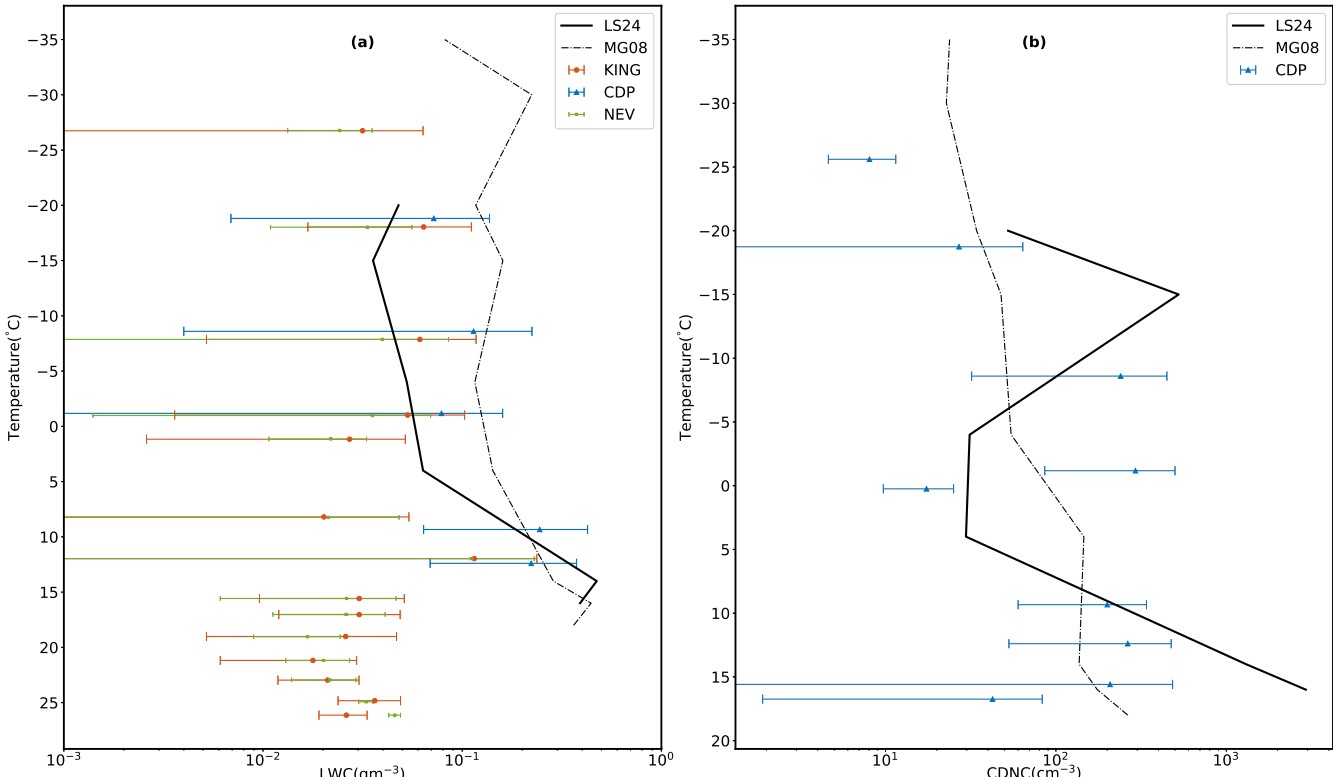

**Figure 2.** Predicted (a) liquid water content (g m$^{-3}$) with CDP, King and Nevzorov probes, (b) cloud droplet number concentrations (cm$^{-3}$) compared with observations from the CDP probe, from the MG08 (dashdotted black line with square) and LS24 (solid black line) simulations. Error bars shown are standard errors of observation samples. The cloud microphysical properties are conditionally averaged over the entire simulation period.

If left uncorrected, the artificial bias from ice shattering on the probe would increase the measured number concentrations of ice particles by an order of magnitude or more, especially at smaller sizes (Korolev et al. 2011). Therefore the observations were corrected with anti-shattering tips and by considering data only for sizes $> 200 \mu m$ and 1 mm. The same size threshold is applied to the simulation results to be consistent. Figure 3a shows agreement of the predicted average of $ni_{200}$ (ice number concentration of particles with sizes $> 200 \mu m$) from LS24 with the aircraft observations, which vary logarithmically in the vertical. The predicted concentration of ice particles has the correct order of magnitude and differs from the observed values by less than the measurement error of the observations, with different probes at any given level differing from each other by almost an order of magnitude. Near the freezing level, $ni_{200}$ predicted by the LS24 run differs by only about $30\%$ from observations.

Figure 3b shows that the prediction of $ni_1$ for precipitation (ice number concentration of particles $> 1$ mm) from LS24 has the same order of magnitude as the aircraft observations in the lower half of the mixed-phase region. Although it is lower by an order of magnitude in the upper half, only one probe (HVPS) for the larger sizes is available and the instrumental error is difficult to quantify (Figure 3b). If there is an uncertainty by an order of magnitude in the measurement of these concentrations,





as noted above for the smaller sizes ($> 0.2$ mm) in view of the spread among various probes, then there is little evidence of any bias with the larger sizes.

Figure 3c shows the ice water content (IWC) predicted by the LS24 run agrees with the observations at most levels, with the
300 correct order of magnitude. However near the $-20$ °C level, the IWC is under-predicted by half an order of magnitude. Overall with the new scheme, the distribution of predicted IWC is much closer to the observations than is the MG08 scheme, as for the filtered ice concentrations noted above.

**Figure 3.** Predicted (a) concentration of ice particles with sizes $> 0.2$mm compared with observations from the 2DC, CIP, HVPS-3 probe and COMB, (b) ice number concentrations of all ice particles with size $> 1$ mm compared with aircraft observations from the HVPS-3 probe and (c) total IWC from the MG08 (dashdotted black line with square) and LS24 (solid black line) simulations. Error bars shown are standard errors of observation samples.



## 4.2  Comparison between the MG08 and LS24 scheme

The MC3E storm was simulated with the original unmodified version of the model, and the run is referred to as the MG08 run.
Accuracy of results is compared between the MG08 and LS24 runs.

The original SCAM6 model (MG08) predicts adequately the timing of the total precipitation peaks at the end of 10 May and 11 May, but under-estimates the intensity of the second (main) peak. The first peak is predicted to be purely convective when according to the detailed simulation it was mostly stratiform (Gupta et al. 2023). The cumulative surface precipitation from MG08 is 60% lower than observations by the end of the simulated period, which is much less accurate than that for LS24
noted above. Generally, only 30% of all precipitation is stratiform for MG08, while detailed high-resolution simulations with AC, comprehensively validated by Gupta et al. (2023), show that this fraction is 80%. Thus LS24 appears to agree better with observations in the cause and intensity of precipitation than does MG08.

Figure 2a shows the LWC predicted by both schemes does not differ so greatly, with a stronger LWC from MG08 by up to a factor of 3 relative to the new scheme (LS24). LS24 is more in agreement with the aircraft observations, though MG08 is
not inconsistent with these. Figure 2b, shows that both runs (MG08, LS24) are similarly in agreement with the observations of CDNC at most levels.

Figure 3a shows that at all levels, $ni_{200}$ values predicted by both schemes mostly agree with the observations, though less so for MG08 than for the new scheme (LS24). Below the $-10$ °C level, the value from MG08 is up to an order of magnitude too low compared with the observations. Generally, the ice concentrations from MG08 are lower than from LS24 by up to an
order of magnitude. Above the homogeneous freezing level ($-36$ °C), $ni_200$ predicted by MG08 is up to an order magnitude lower than the LS24 run. Figure 3b shows that the precipitation concentration, $ni_1$, predicted by the MG08 run, is at least an order of magnitude lower than observations at all flight levels, being lower than LS24 by at least half an order of magnitude at most subzero levels.

Figure 3c shows that the original MG08 run under-predicts IWC by about half an order of magnitude throughout the mixed-
phase region compared to observations. At most subzero levels, it is less than LS24 by up to an order of magnitude.

The MG08 run predicts net radiative fluxes slightly less accurately than the LS24 run. At the TOA the net shortwave and longwave fluxes are 30% and 15% too low respectively for MG08 compared to observations.



# 5 Results from Sensitivity tests

**Table 5.** List of the sensitivity simulations

| Name of simulation | Description |
|---|---|
| Control | Control simulation with the LS24 scheme discussed here |
| no-breakup | Fragmentation during ice-ice collisions process is inactive in the LS24 run |
| no-rfz | Fragmentation during raindrop freezing process is inactive in the LS24 run |
| no-HM | HM process is inactive in the LS24 run |
| no-SIP | All three SIP mechanisms are inactive in the LS24 run |
| no-HOMO | Homogeneous freezing is inactive LS24 run |
| no-HOMO + no-SIP | Homogeneous freezing and All three SIP mechanisms are inactive are inactive LS24 run |
| no-HOMO STRATO | Homogeneous freezing is inactive in stratiform clouds LS24 run |
| high-INP | Active IN concentrations increased by a factor of 100 relative to the LS24 run in stratiform clouds |

Sensitivity simulations were conducted to systematically examine the influence of different microphysical pathways of ice
initiation and the role of environmental concentrations of active ice nuclei (IN) originating from solid aerosols such as mineral
dust. Starting from the baseline LS24 run (Sec. 4.1), a suite of perturbation experiments was performed, each designed to isolate
the effect of a specific microphysical or environmental modification. These included the following runs: (i) 'no-breakup', where
the ice–ice collisional breakup process was prohibited; (ii) 'no-rfz', in which the rime-splintering process was switched off;
(iii) 'no-HM', which removed all Hallett–Mossop ice multiplication effects; (iv) 'no-SIP', where secondary ice production by
other pathways was switched off; (v) 'no-HOMO', where homogeneous freezing was deactivated; (vi) 'no-HOMO + no-SIP',
excluding both homogeneous freezing and secondary ice production; (vii) 'no-HOMO STRATO', removing homogeneous
freezing exclusively in the stratospheric layers; and (viii) 'high-INP', with elevated ambient concentrations of ice-nucleating
particles to mimic enhanced dust loading. Comparison of these targeted perturbation simulations with the control run (LS24)
allows for a detailed attribution of changes in cloud microphysics, radiative fluxes, and precipitation characteristics to specific
ice initiation mechanisms and ice-nucleating aerosol conditions. Full descriptions of each configuration are provided in Table 5.




**Table 6.** Unconditional average of the radiative fluxes for the simulation period from 00 UTC 10 May 2011 to 00 UTC 13 May 2011 for the sensitivity simulations. Positive values of SW and LW flux signify radiations propagating downwards and upwards, respectively, into the climate system.

| Radiation fluxes (W/m²) | Net Shortwave (SW) radiative flux at TOA | Net Longwave (LW) radiative flux at TOA | Net Shortwave radiative flux (SW) at surface | Net Longwave (LW) radiative flux at surface |
|---|---|---|---|---|
| Control | 266.68 | 196.55 | 149.44 | 49.11 |
| no-brk | 262.78 | 205.47 | 150.78 | 52.84 |
| no-rfz | 258.36 | 198.59 | 142.79 | 50.80 |
| no-HM | 269.72 | 196.72 | 151.45 | 49.42 |
| no-SIP | 263.52 | 187.64 | 148.75 | 50.15 |
| no-HOMO | 254.86 | 203.76 | 142.05 | 53.72 |
| no-HOMO+no-SIP | 265.81 | 200.38 | 154.33 | 53.80 |
| no-HOMO STRATO | 272.47 | 197.95 | 155.81 | 50.02 |
| high-INP | 259.55 | 179.13 | 141.03 | 46.33 |

## 5.1 Ice initiation pathways

The simulations show the various sensitivities to different pathways of ice initiation. The mechanisms were prohibited as discussed in Table 5 in both the stratiform and convective microphysics schemes.

Figure 4a shows a major impact from homogeneous freezing on the ice concentration in stratiform regions simulated. Without homogeneous aerosol freezing in the large-scale clouds (no-HOMO STRATO case), the ice concentration is reduced by two orders of magnitude in the mixed phase region of temperature (0 to -36 °C ) and by even more aloft in cirriform clouds. Further removal of all other homogeneous freezing causes only a slight reduction by up to 30% relative to the no-HOMO STRATO case. Regarding SIP, the no-SIP run reveals little systematic impact on stratiform ice concentration from all SIP in the control run, which is explicable in terms of the prevalence of homogeneous ice throughout all sub-zero levels. Switching off individually each SIP mechanism perturbs the ice concentration by up to ±50%, due to alteration of precipitation amounts aloft and supercooled cloud liquid properties, affecting homogeneous freezing in a non-linear way.

Figure 4b shows a marked reduction of IWC by about an order of magnitude in the upper half of the mixed-phase region when homogeneous aerosol freezing in the large scale cloud scheme is prohibited. There is a similar reduction in the cirriform cloud at upper levels. This is all consistent with the decrease in ice number concentration noted above. The complete removal of all other homogeneous freezing (no-HOMO) further decreases IWC slightly (by 20%) relative to no-HOMO STRATO, with a comparatively small incremental change. The no-SIP run results in minor changes to the IWC profile relative to the control, reaffirming the dominance of homogeneous freezing in driving ice mass in the simulations. Switching off individual





SIP processes introduces variations in IWC of up to $\pm 40\%$, attributed to their perturbation of mixed-phase cloud microphysics, especially through indirect effects on alterations of vapor growth, riming, and sedimentation of ice.

Prohibiting all SIP reduces the snow number concentration by up to a factor of two in the mixed phase region (Figure 4c) relative to the control. Excluding breakup in ice-ice collisions reduces the snow number concentration by half an order of magnitude in the lower half of the mixed phase region. The corresponding changes in the snow mass content are minimal.

Figures 5a and b show how the cloud liquid properties are influenced by SIP mechanisms and homogeneous freezing. The no-SIP run shows an increase in the supercooled LWC throughout the mixed phase region by $10\%$ relative to the control,
because the growth of fewer ice particles causes less depletion of liquid by evaporation and riming. For similar reasons, in the lower half of the mixed phase region of the cloud, prohibiting all homogeneous freezing boosts the LWC by about half an order of magnitude relative to the control. The cloud droplet number concentration is mostly increased by excluding either homogeneous freezing or SIP, because the descent of homogeneous ice into the mixed-phase region creates more subsaturation with respect to water, evaporating supercooled cloud-liquid. Figure 5c,d shows that the mass and number concentrations of
rain are perturbed but not in a systematic manner at all levels by the exclusion of various SIP mechanisms.

Figure 6a, b shows that the two major convective peaks seen in the observations (10 May, 21:00 and 11 May, 21:00) are predicted to be boosted by about 20% from the SIP being prohibited, but the stratiform precipitation following the major peak on the final day is slightly weakened (by 20%). Switching off all the homogeneous freezing drastically weakened the stratiform precipitation on the final day by a factor of about 2, similarly diminishing the cumulative surface precipitation. The no-breakup
case run had a cumulative surface precipitation very similar to the no-SIP run, indicating that the breakup in ice-ice collisions dominates the overall SIP impact on precipitation.

Figure 6 c, d shows the impact on the top of the atmosphere radiation, also detailed in table 6. The no-SIP run has a TOA Net Shortwave flux entering the atmospheric column that is 3 W/m$^2$ lower than control due to more cloud condensate aloft at sub-zero levels with a corresponding reduction in the outgoing long-range radiation, relative to control. Without homogeneous
freezing, there is a stronger reduction (by 12 W/m$^2$) in SW flux entering the climate system than without SIP due to more cloud condensate aloft. However, at upper levels there is less homogeneous ice and more emission of longwave radiation to space from warmer average emitting level when homogeneous freezing is excluded.







**Figure 4.** Predicted (a) Ice particle number concentration ($L^{-1}$), (b) IWC ($gm^{-3}$), (c) Snow number concentration ($L^{-1}$), (d) Snow mass content ($gm^{-3}$). The cloud microphysical properties are conditionally averaged ( over stratiform cloud regions only) over the entire simulation period.





**Figure 5.** Predicted (a) Liquid Water Content (g m$^{-3}$), (b) cloud droplet concentration, CDNC (g cm$^{-3}$), (c) Ice water content (g m$^{-3}$), (d) CDNC (cm$^{-3}$), (c) snow mass mixing ratio, (d) rain number concentrations (L$^{-1}$). The cloud microphysical properties are conditionally averaged (over stratiform cloud regions only) over the entire simulation period.



**Figure 6.** Predicted (a) surface precipitation rate (mm/hr), (b) accummulated surface precipitation (mm), (c) net shortwave flux at TOA (W/m$^2$, positive downward), and (d) net longwave flux at TOA (W/m$^2$, positive upward). In (a) and (b), only the stratiform components of precipitation are shown.





## 5.2   Environmental aerosol conditions affecting ice nucleation

Figure 7a illustrates the response of ice number concentration in the sensitivity experiment where the concentration of ice
nucleating particles (INPs) in the environmental aerosol population is increased by a factor of 100 at all levels. Despite already
high baseline loadings of INPs, the Figure reveals that the ice number concentration exhibits more than an order-of-magnitude
variation across the temperature range. In the control simulation at subzero levels, homogeneous freezing is the dominant mech-
anism for ice formation overall, since it occurs at temperatures colder than about -36 degC, and humidities approaching water
saturation (high supersaturations with respect to ice). This leads to a high number of ice crystals aloft at upper levels which
are then downwelled somehow. In contrast, the high-INP case (100 × INP numbers) activates heterogeneous nucleation much
earlier during (e.g., large-scale) ascent—at warmer temperatures and lower supersaturation. This early onset of ice formation
limits the buildup of supersaturation necessary for homogeneous freezing, thereby suppressing it. As a result, the homoge-
neous nucleation pathway is effectively "switched off" in the high-INP case, leading to a notable reduction in total ice number
concentration, as is evident in Figure 7. Thus, the model represents the known competition between homogeneous aerosol
freezing and heterogeneous ice nucleation in cirriform clouds (e.g. Kärcher and Lohmann (2002)). The early consumption of
water vapor by heterogeneous nucleation prevents further crystal formation and shifts the microphysical regime.

Moreover, with fewer ice crystals forming overall in the high-INP case, the competition for water vapor during growth is
reduced. This enables existing crystals to grow larger, contributing to increased snow production. The enhanced snow growth
can lead to greater meltwater generation and subsequently more rainfall through the ice crystal process of precipitation (or
'cold-rain process'), consistent with findings reported by Gupta et al. (2023). Finally, Figure 7 also reflects a shift in the
equilibrium saturation conditions—ice formation occurs at a lower saturation threshold in the high-INP scenario

Figure 7b shows that the ice water content (IWC, $gm^{-3}$) follows a pattern consistent with the response of ice number
concentration to increased INP loadings under the 100× INP sensitivity scenario. In the control simulation, where homogeneous
freezing dominates at lower temperatures and high supersaturation, the rapid formation of numerous ice crystals leads to
elevated IWC values. In contrast, in the high-INP case, earlier initiation of heterogeneous nucleation results in fewer crystals
forming, thereby lowering IWC in regions that would otherwise favor homogeneous freezing. However, the larger size of ice
particles in the high-INP case noted above partially compensates and maintains moderate IWC values across a portion of the
temperature range. This balance between fewer but larger crystals shapes the overall IWC response and underlines a shift in
the ice growth regime.

Figure 7c also illustrates the snow number concentration ($L^{-1}$), providing insight into how ice particles evolve into precipitation-
sized snow. In the control case, where homogeneous nucleation yields a high number of small ice crystals, subsequent growth
processes (e.g., riming and aggregation) produce a relatively large number of snow particles. In the high-INP simulation, how-
ever, the fewer initial crystals noted above cause the downstream formation of snow particles, leading to lower snow number
concentrations. The reduced competition for water vapor, and the greater abundance of supercooled cloud-liquid from fewer
crystals in the large-scale cloud (Figure 8), in this scenario favors the growth of fewer, but larger, snow particles. This further
highlights the shift in microphysical pathways due to increased INP concentrations.





The snow mass content ($\mathrm{gm}^{-3}$) in Figure 7d shows that, while the snow number concentration is reduced in the high-INP scenario relative to the control, the snow mass content does not decline proportionally. This is because the reduced number of snow particles grow more efficiently, reaching larger sizes as with the ice crystals. In some temperature regimes, the snow mass mixing ratio is even enhanced relative to the control case, signifying more efficient precipitation development. This outcome supports the interpretation that the high-INP regime promotes a shift toward fewer but more massive snow particles, enhancing cold-rain production from melting of ice precipitation (Figure 8), in line with the findings of Gupta et al. (2023).

Figure 9a shows that in the control case, where homogeneous nucleation dominates, surface precipitation occurs in intermittent bursts with moderate peak intensities. By contrast, the high-INP case has more frequent and stronger peaks in surface precipitation rate. This outcome arises from earlier activation of heterogeneous nucleation, which limits homogeneous ice formation but favors the rapid growth of fewer ice crystals into precipitation-sized hydrometeors. The cumulative precipitation (mm) in Figure 9b further highlights this difference. Both simulations initiate precipitation at nearly the same time, yet the high-INP case produces considerably more accumulated surface precipitation. This is consistent with a microphysical regime where fewer but larger ice and snow particles grow more efficiently, ultimately enhancing cold-rain production through melting, as also noted in Gupta et al. (2023) Gupta et al. (2023). Figures 9c–d show the radiative implications of these changes at the top of the atmosphere (TOA). The net shortwave flux (Figure 9c) is slightly reduced in the high-INP case, reflecting enhanced cloud shading due to more persistent condensate and precipitation. Meanwhile, the net longwave flux (Figure 9d) reveals systematically lower outgoing fluxes at TOA for the high-INP simulation, consistent with deeper, longer-lived cloud layers that trap infrared radiation, partially re-emitting it at a colder temperature aloft to space. Taken together, Figure 9 demonstrates that the increase in INP concentrations not only modifies the microphysical pathways of ice initiation—shifting from homogeneous to heterogeneous nucleation—but also amplifies precipitation production and alters the radiation budget at the TOA.

In summary, the higher numbers of INPs act to shift the dominant pathway of ice initiation from homogeneous to heterogeneous ice nucleation in the simulations, as expected. This occurs by their effect on lowering the humidity aloft. Such changes in nucleation pathways not only influence the number of hydrometeors but also their mass, with significant implications for precipitation.





**Figure 7.** Predicted (a) ice particle number concentration ($L^{-1}$), (b) IWC ($gm^{-3}$), (c) snow number concentration ($L^{-1}$), and (d) snow mass content ($g\ m^{-3}$). The cloud microphysical properties are conditionally averaged (over stratiform cloud regions only) over the entire simulation period.



**Figure 8.** Predictions from the high-INP case for (a) liquid water content (g m$^{-3}$), (b) cloud droplet concentration, CDNC (cm$^{-3}$), (c) rain mass content (g m$^{-3}$), and (d) rain number concentrations (L$^{-1}$). The cloud microphysical properties are conditionally averaged (over stratiform cloud regions only) over the entire simulation period.





**Figure 9.** Prediction of surface stratiform precipitation and radiative fluxes at TOA for the high-INP case, plotted as in Fig. 6.





## 6   Conclusions

In this study, the large-scale cloud microphysics scheme MG08 has been substantially modified through the introduction of physically-based representations for key microphysical processes, as described in Section 2.2. This updated scheme—referred to as LS25—was implemented within SCAM6 and used to simulate the MC3E storm, with a focus on understanding the role 445 of homogeneous nucleation and secondary ice production (SIP) mechanisms in shaping mixed-phase cloud microphysics and associated storm properties.

Key conclusions are as follows:

1. The new scheme improves the accuracy of the prediction for the control run of the MC3E storm:

    – There is a 10 % slight over-prediction of cumulative surface precipitation over the entire simulationm with the new 450 scheme, compared with an under-prediction by 60% with the original scheme by MG08. Moreover, the surface precipitation is predicted for the right macrophysical reasons, with 65 % coming from stratiform precipitation compared with 30% for the original scheme over the entire simulated period. The corresponding fraction is about 80% from our detailed high-resolution simulations (AC), (Gupta et al. 2023).

    – Predictions of concentrations of ice concentrations and related supercooled LWC are also improved.

– Prediction of SW radiative fluxes at TOA and surface are improved (low bias at TOA reduced from about 30% to 20%) due to reduction of the over-estimate of cloud condensate amount aloft. Little improvement occurs in the LW fluxes, however.

2. In the control run, supercooled cloud droplets in stratiform clouds are largely depleted through accretion onto rain and ice rather than undergoing homogeneous freezing. Sensitivity experiments confirm that homogeneous freezing of 460 solute aerosols, rather than supercooled cloud-droplets, plays a dominant role in controlling ice number concentrations in large-scale regions at upper levels when the humidity approaches water saturation, as expected from high-resolution cloud simulations (Phillips et al. 2007). In the simulation, this homogeneous ice is downwelled through the mixed-phase region, albeit arguably too far down.

3. Prohibiting homogeneous freezing in the large-scale (stratiform) cloud regime leads to a reduction by up to two orders 465 of magnitude in ice number concentration, especially within the mixed-phase temperature range (0 to –36,°C). This also significantly reduces the IWC, with further suppression when all homogeneous pathways are disabled. SIP processes, while not dominant in this regime, contribute up to $\pm 50\%$ variations in ice concentration and IWC through their nonlinear interactions with cloud liquid and precipitation processes.

4. Increasing environmental concentrations of active INPs by a factor of 100 triggers earlier heterogeneous nucleation 470 during ascent, which inhibits the buildup of supersaturation required for homogeneous freezing. This transition reduces total ice and snow number concentrations but allows larger ice particles to form due to reduced vapor competition and



more supercooled cloud-liquid for riming. The resulting microphysical shift enhances snow mass mixing ratios and is consistent with an increased cold-rain process.

5. Radiative impacts are also significant: sensitivity tests prohibiting either homogeneous freezing or SIP show reductions in both shortwave (net incoming) and longwave (outgoing) net radiative fluxes at the top of the atmosphere (TOA) due to more cloud condensate aloft and increased upper-level cold hydrometeors. These findings demonstrate the strong feedback between microphysical pathways and cloud-radiative interactions.

Regarding point 1 here, the updated LS25 scheme improves SCAM6's ability to represent stratiform cloud microphysics. Notably, the improved treatment of convective detrainment and large-particle ice number concentrations ($> 1$ mm) leads to a more realistic stratiform-to-convective precipitation ratio. This aligns better with observations and addresses known biases, such as the historical $\sim$70% underestimation of stratiform precipitation in SCAM simulations. Consistent with Gupta et al. (2023), who found that about 80% of surface precipitation during the MC3E storm came from stratiform clouds, our results confirm the critical importance of accurately representing ice initiation mechanisms in large-scale clouds to capture precipitation and cloud radiative effects.

To conclude, the sensitivity of the storm system to aerosol loading is nonlinear and complex. Aerosol-induced changes to the environmental INP levels significantly modulate the formation, growth, and sedimentation of ice-phase hydrometeors by altering the balance between heterogeneous and homogeneous nucleation. These changes ultimately impact precipitation formation pathways, cloud longevity, and radiation budgets. The present paper shows that conventional global models with upgraded treatment of cloud microphysics can capture much of this complexity in the aerosol-cloud linkage.

*Acknowledgements.* This work was chiefly supported by an award (2018-01795) from FORMAS to VTJP, regarding the modelling of ice initiation in clouds and climate. This grant supported CSP, who created the stratiform codes and finished the paper. VTJP was also supported by an award (2021-01463) to VTJP from the Swedish Research Council for Sustainable Development (FORMAS). SP was supported by an award to VTJP from Vinnova (2020-03406). The authors are grateful to Arti Jadav, who wrote the first draft of the paper and developed the convective scheme in the SCAM simulations. We would like to express our appreciation to the contributors of the Community Earth System Model. The authors acknowledge John Truesdale for offering valuable guidance and assistance concerning the CESM model.

**Appendix A:**

Table A1: List of Symbols

| Symbol | Description | Units and/or Value |
|--------|-------------|--------------------|
| $D$ | Diameter of drop just before freezing | $m$ |
| | | Continued on next page |





**Table A1 – continued from previous page**

| Symbol | Description | Units and/or Value |
|---|---|---|
| $D_i$ | Maximum dimension of crystals | $m$ |
| $n(D_x)\, dD_x$ | Number concentration of the hydrometer in the size range $dD_x$ | m$^{-3}$ |
| $D_{pmax}$ | Droplet diameter at maximum supersaturation in the $j^{th}$ bin for $i^{th}$ aerosol | m |
| $D_x$ | Equivalent spherical diameter of the cloud micro-physical species corresponding to subscript $x$ | m |
| $E_c$ | Collision efficiency | - |
| $l = 1, 2$ | Indicates which particle is fragile | - |
| $(m_1\, m_2)$ | Mass of colliding particles | kg |
| $(m_b, m_t)$ | Initial mass of big and tiny fragments | kg |
| $m_x(i)$ | Mass mixing ratio of interacting particle $x$ in $i^{th}$ size bin | kg kg$^{-1}$ |
| $m_y(j)$ | Mass mixing ratio of interacting particle $y$ in $j^{th}$ size bin | kg kg$^{-1}$ |
| $n_x(i)$ | Number concentration of interacting particle $x$ in $i^{th}$ size bin | kg$^{-1}$ |
| $n_y(j)$ | Number concentration of interacting particle $y$ in $j^{th}$ size bin | kg$^{-1}$ |
| $m_{x,1}$ | Mass concentration of the hydrometer in first size bin | kg kg$^{-1}$ |
| $N$ | Number of secondary ice particles per frozen drop, excluding the parent drop | - |
| $N_{aerosol}(i,j)$ | Number concentration of $i^{th}$ aerosol in the $j^{th}$ size bin in the solid aerosol group (sulphate in Mode 1 and Mode 2, secondary organic matter, sea salt) | kg$^{-1}$ |
| $N_B$ | Number of big secondary ice particles per frozen drop, excluding the parent drop | - |
| $N_T$ | Number of tiny secondary ice particles per frozen drop | - |
| | Continued on next page | |




**Table A1 – continued from previous page**

| Symbol | Description | Units and/or Value |
|---|---|---|
| $n_{x,0}$ | Intercept of the hydrometeors corresponding to subscript $x$ | - |
| $n_{X',a}$ | Number of aerosols lost by ice nucleation in group $X'$ | kg$^{-1}$ |
| $n_{IN,X'}$ | Contribution to $n_{IN}$ from aerosol group $X'$ | kg$^{-1}$ |
| $n_{IN,X',a}$ | Number of aerosols in aerosol group $X'$ lost by ice nucleation | kg$^{-1}$ |
| $p_w$ | Shape parameter of the cloud microphysical species corresponding to subscript $x$ | - |
| $q_v$ | Vapour mass mixing ratio | kg kg$^{-1}$ |
| $q_c$ | Cloud droplet mass mixing ratio | kg kg$^{-1}$ |
| $q_i$ | Cloud-ice mass mixing ratio | kg kg$^{-1}$ |
| $q_s$ | Snow mass mixing ratio | kg kg$^{-1}$ |
| $q_r$ | Rain mass mixing ratio | kg kg$^{-1}$ |
| $q_g$ | Graupel/hail mass mixing ratio | kg kg$^{-1}$ |
| $v\,v_i)$ | Fall velocity of drops and crystals | ms$^{-1}$ |
| $(v_1\,v_2)$ | Fall velocity of colliding particles | ms$^{-1}$ |
| $\chi_{x,y}$ | Collection kernel for the interacting particles $x$ and $y$ | |
| $(\delta N_1, \delta N_2)$ | Concentrations of pair of colliding particles in size ranges $(\delta r_1, \delta r_2)$ | m$^{-3}$ |
| $\Delta n_{x,y}$ | Change in number concentration for particle $x$ collecting $y$ | kg kg$^{-1}$ |
| $\Delta q_{x,y}$ | Change in mass mixing ratio for particle $x$ collecting $y$ | kg kg$^{-1}$ |
| $\Delta n_c$ | Number of cloud droplets generated | kg$^{-1}$ |
| $\Delta n_i$ | Number of cloud-ice generated | kg$^{-1}$ |
| $\Delta t$ | Time step of the global model | 1200 s |
| $\lambda_x$ | Slope parameter of cloud microphysical properties corresponding to subscript $x$ | - |
| $\mathcal{N}$ | Number of fragments per collision | - |



**Table A1 – continued from previous page**

| Symbol | Description | Units and/or Value |
|---|---|---|
| $\rho$ | Density of air | kg m$^{-3}$ |
| $\rho_w$ | Density of water | kg m$^{-3}$ |
| $\tilde{n}$ | Number concentration of drops | m$^{-3}$ |
| $\tilde{n}_i$ | Number concentration of ice particles | m$^{-3}$ |
| $\zeta$ | Ratio of initial fragment mass to mass of parent particle (more fragile of the colliding particle) | - |



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
