# Peer review of "A modified parameterization of stratiform cloud microphysics for the Community Earth System Model"

_EGUsphere, 2025_

## Referee Comment (RC1)

Review for "A modified parameterization of stratiform cloud microphysics for the Community Earth System Model"

The study (Pant et al.) discusses development of the stratiform cloud microphysics scheme to represent large-scale clouds in climate models. This work shows that an improved treatment of the interactions between aerosols and large-scale stratiform clouds can be achieved by better representing the secondary ice production in the model. In doing so, authors were able to obtain more realistic predictions of cloud properties using the Single-Column Atmospheric Model (SCAM6), which they validated using the observations from the MC3E campaign over Southern Great Plains.

Ice processes in climate models are uncertain. Through this work, the authors show that model predictions of cloud properties can be sensitive to both primary and secondary ice production processes, and hence, these processes could play a vital role in improving the prediction and evaluation of clouds in models. I recommend its publication in "Atmospheric Chemistry and Physics" after the following comments are addressed.

**Major comments:**

Line 100: The subsection "Treatment of aerosols" can be improved. Where were the biological and non-biological aerosols derived from? Are any of these sources biomass burning? CAM6 uses MAM4 (Liu et al. 2016) for the treatment of aerosol species and processes. Was this entirely replaced by the scheme provided by Phillips et al. (2009) or only some processes? What are the seven modes in the aerosol model called? Please provide additional information.

Line 271: It looks like the precipitation between May 12th and 13th is being largely overpredicted by LS24 (Figure 1a). What processes are causing that? It also seems like the initiation of the precipitation is off by a few hours in all the simulations. Do the authors know a possible explanation for that? It would be useful to add some discussion for that in the text.

Figure 4: It appears that the "no-SIP" case does not really cause any marked difference in the ice-phase cloud properties relative to the control simulation (Fig. 4). Is this because the model is not sensitive to SIP processes or is SIP physically not very important for ice-phase and mixed-phase processes? Can the authors please provide some explanation for that?

**Minor comments:**

Line 71: The double-moment stratiform microphysics options in CAM6 are MG1 (Morrison and Gettelman, 2008) and MG2 (Gettelman et al. 2015), and not Morrison and Grabowski (2008). Please check the citation.

Line 76: "were omitted Liu and Penner (2005)". Typo?

Line 76: Is the citation correct? Liu and Penner (2005) studied Primary Ice Production mechanisms via homogenous and heterogeneous nucleation modes. As far as I know, they did not discuss any mechanism for SIP. Please check.

Table 1: Why are there different sigmas for each species? Do these sigmas correspond to different aerosol modes? Also, do the geometric mean diameters correspond to the sizes at which these aerosols are being emitted?

Eq. 3: Please cite Morrison et al. (2005) for the "lambda x" parameter.

Eq. 5: How do you obtain nx,0?

Line 138: Do you mean "in-cloud droplet activation follows Abdul-Razzak and Ghan (2000)"? Wouldn't k-Kohler theory be used to only estimate the activation supersaturation? In-cloud activation would require calculating the maximum ambient supersaturation for activation, which we cannot get simply from the k-Kohler theory. Also, is there a reason why Ming et al. (2006) was not used for in-cloud activation?

Line 255: Were the temperature and wind fields also relaxed to the observations for the SCAM6 simulations? If these are atmospheric isotherms, I suggest the plots to be made with respect to isobars or vertical height.

Line 263: What does "LS25" stand for?

Line 270: Figure 1(a) and (b) have not been separately referenced when being discussed.

Line 271: What does "LS24" stand for?

Figure 2: What are the temperatures on the y-axis? All the text and marker sizes in the plot can be increased for better readability. Furthermore, adding standard errors/standard deviations for the simulated LWC and CDNC would also be useful.

Figure 2b: It looks like the variability in CDNC from LS24 is much larger than what is being suggested by the observations. Similar variability is not very evident in the LWC data. Can the authors comment on that?

Line 320: The limit for the homogenous freezing is mentioned as -36C, but In section 2.2.7, it is mentioned as -35C. Is it the former or latter? I would suspect 1C difference at these temperatures should substantially change the INPs. Can the authors please confirm?

Line 324: What is classified as the "mixed-phase region"?

Table 4 and 6: What is an "Unconditional average"?

Table 4 and 6: How do the shortwave and longwave cloud forcing values change between all these experiments? Have they been evaluated?

Line 388: "deg". Typo.

Line 391: I am slightly confused about the discussion regarding the ice nucleation and the distribution of the ice particle number concentrations. Homogenous freezing would be "effectively turned off" above the -35C mark where the ice particles would be greater in number in the control simulations. But shouldn't the aerosol-induced freezing result in a larger ice number concentration at warmer temperatures? Can the authors please explain.

Line 444: I am not sure which scheme is LS25, the discussion in the manuscript only mentions LS24 throughout the writeup

---

## Referee Comment (RC2)

**Review**

*A modified parameterization of stratiform cloud microphysics for the Community Earth System Model.*

**Recommendation:** Accept with revisions.

The paper illustrates the effects of modifications to the microphysics parameterization in the Community Atmosphere Model, version 6 (CAM6). The modifications deal with species representation; droplet activation; additional mechanisms for secondary ice production (SIP); and emulations from bin microphysics for aggregation, accretion, and riming. The paper also diagnoses the impact of homogeneous freezing and SIP and analyzes the effect of large increases in ice nucleating particles (INP). Only one case is analyzed, using SCAM6, the single column model for CAM6.

Overall, the paper provides important documentation of the behavior of advances in cloud microphysics in atmospheric models and warrants publication. The mechanism denial experiments for homogeneous freezing and SIP do not align well with the modifications to the microphysics but are nonetheless of interest, as are the sensitivity experiments for INP.

Revisions are suggested in the following.

**Major Revisions**

1. ll. 93, 478: The statement should describe more clearly the Jadav et al. (2025) convective parameterization. Is it the Zhang-McFarlane (ZM) parameterization in SCAM6 (Gettelman et al., 2019), but with only changes in microphysics, or are there also changes in other basic attributes like its plume model, closure, and entrainment-detrainment assumptions? The MG08 simulations presumably use the convection parameterization in the default SCAM6. If Jadav involves changing more than just the convective microphysics in ZM, the differences between the MG08 and LS24 (or LS25, hereafter LS) simulations are not just due to the microphysics modifications but also other differences in the treatment of convection. If this is the case, the paper should distinguish between the effects of microphysics modification and changes in treatment of convection not related to microphysics. L. 478 refers to "improved treatment of convective detrainment." Is the source of

this improvement just microphysical, or does it also involve other changes introduced in the Jadav convective parameterization?

2. Fig. 1-3: In general, the observational uncertainties are large. In some cases, e.g. Fig. 3a, it's difficult to confidently choose between the two parameterizations regarding possible consistency with observations. On Fig. 3b, MG08 correlates better than LS with CIP but is more biased. For Figs. 1-3, I recommend adding metrics to the revised text (bias, RMSE, correlation coefficients) for MG08 and LS relative to observations.

3. Only one case in a single field campaign is presented. The authors should note this limitation and offer any insights possible on how general their results are.

**Minor Revisions**

1. CESM would more accurately be referred to throughout the paper as CAM6. CESM is generally taken to refer to the fully coupled Earth system model, of which CAM6 is a component. The paper is restricted to a single-column application of CAM6 with no coupled or global results.

2. Consider re-titling to avoid "for CESM." The parameterization has not at present been included in CESM, and the parameterization is sufficiently general that it could be used in other general circulation models for climate or numerical weather prediction. A revised title could be something like "Single-column evaluation and diagnosis of modified cloud microphysics in the Community Atmosphere Model, version 6".

3. l. 460: The paper does not show the relative roles of homogeneous droplet freezing versus homogeneous aerosol freezing. Can this statement be supported further?

4. On l. 251, specify whether any nudging to observations was done when applying large-scale tendencies in SCAM6.

5. ll. 417, 421, 470: Text states snow number concentration is reduced in high-INP, but for almost all temperatures warmer than $-36^{\mathrm{O}}$C Fig. 3c shows increased snow number concentration at high-INP.

6. ll. 278, 326, 327: The LS radiation predictions are more accurate for SW but not LW.

7. l. 86: Define acronym AC here, its first occurrence, rather than later at l. 95.

8. The paper mixes use of "LS24" and "LS25." This should be consistent throughout.

9. It's unclear whether the "MG08" model against which "LS" is compared used the Jadav et al. (2025) convective microphysics or the convective microphysics cited in Gettelman et al. (2019). If the latter, titling and other characterizations of the microphysics should not be modified by "stratiform," as both convective and stratiform microphysics would be modified.

10. The Ming et al. (2006) activation scheme depends on vertical velocity. The grid-mean vertical velocity is not representative of subgrid variability in vertical velocity, e.g., at cloud bases, cf. Golaz et al. (2011, *J. Clim.*). How are vertical velocities specified in your application of Ming et al. (2006)?

11. Are aerosols also depleted by dry deposition? Mention in text, if so.

12. Table 1, Mineral dust geometric mean diameter: Should second number be smaller than first?

13. l. 112: The slope parameter $p_x$ would be better defined here rather than later at l. 124. Consider moving the definitions for the distributions to this part of the paper, rather than including them later in the text. Mention early in the text, when symbols first appear, that you have provided Table A1 with a complete list of symbols.

14. l. 131: Ming et al. (2006) also links the droplet number concentration to vertical velocity.

15. Fig. 5 legend: errors regarding item (c)

16. ll. 369-370: Fig. 5(d) shows noticeable impacts from excluding breakup and homogeneous freezing. Also, at temperatures just below freezing, excluding SIP cancels much of the effect of excluding homogeneous freezing. Can this be commented upon?

- l. 72: "Grabowski" $->$ "Gettelman"?
- l. 320: subscript error
- l. 338: "stratospheric" $->$ "stratiform"?

---

## Author Comment (AC1)

26 January 2026
To
The Editor-in-Chief
Atmospheric Chemistry and Physics (ACP)

**Subject: Resubmission of revised manuscript: egusphere-2025-4740, "A modified stratiform cloud microphysics parameterization: evaluation using the Community Atmosphere Model version 6 single-column model"**

Dear Madam/Sir,

We are pleased to resubmit our revised manuscript entitled "*A modified stratiform cloud microphysics parameterization: evaluation using the Community Atmosphere Model version 6 single-column model*" (Manuscript ID: **egusphere-2025-4740**) for reconsideration in *Atmospheric Chemistry and Physics*. We sincerely thank you and the reviewers for the time, effort, and constructive comments provided during the review process. The reviewers' insights have been invaluable and have significantly helped us improve the scientific clarity, technical rigor, and overall presentation of the manuscript.

In response to the reviewers' comments, we have carried out a thorough revision of the manuscript. The major improvements and modifications include, but are not limited to, the following:

1. Clarification and expansion of the treatment of aerosols, including detailed descriptions of aerosol sources, modal structure, and their relationship to the CAM6 aerosol framework, with explicit discussion of biological, non-biological, and biomass-burning contributions.

2. Improved documentation of the stratiform and convective microphysics coupling, clearly distinguishing modifications to microphysical processes from the unchanged dynamical and closure assumptions of the convection scheme.

3. Additional discussion explaining the physical dominance of homogeneous freezing relative to secondary ice production (SIP) for the simulated MC3E case, thereby clarifying the sensitivity experiment outcomes.

4. Inclusion of quantitative metrics for evaluation of biases to complement the qualitative comparison with aircraft, satellite, and ground-based observations.

5. Improved clarity and consistency throughout the manuscript, including corrected citations, unified terminology (e.g., LS24), clearer definitions of the mixed-phase region, and resolution of typographical and figure-related issues.

6. Revised title.

A detailed, point-by-point response to all reviewer comments has been prepared and is submitted alongside the revised manuscript. We believe that the revisions have substantially strengthened the manuscript and improved its suitability for publication in *Atmospheric Chemistry and Physics*. The study provides a physically consistent and observationally evaluated advancement in the representation of stratiform cloud microphysics, with clear relevance to the ACP readership interested in cloud–aerosol interactions and climate-model development.
We confirm that this manuscript is original, has not been published previously, and is not under consideration for publication elsewhere.
The resubmission includes the following files:

- A detailed, point-by-point response to the reviewers' comments.

- A marked-up version of the revised manuscript highlighting changes (colored in blue).

- A clean version of the revised manuscript incorporating all revisions.

We sincerely appreciate your consideration of our revised submission. We look forward to your response and would be happy to address any further comments or suggestions.

Sincerely,
Chandra Shekhar Pant

Department of Hydro and Renewable Energy
Indian Institute of Technology Roorkee
Roorkee, India
Email: csp@hre.iitr.ac.in

---

## Author Comment (AC3)

**Reviewer # 2**

*The study (Pant et al.) discusses development of the stratiform cloud microphysics scheme to represent large-scale clouds in climate models. This work shows that an improved treatment of the interactions between aerosols and large-scale stratiform clouds can be achieved by better representing the secondary ice production in the model. In doing so, authors were able to obtain more realistic predictions of cloud properties using the Single-Column Atmospheric Model (SCAM6), which they validated using the observations from the MC3E campaign over Southern Great Plains.*

*Ice processes in climate models are uncertain. Through this work, the authors show that model predictions of cloud properties can be sensitive to both primary and secondary ice production processes, and hence, these processes could play a vital role in improving the prediction and evaluation of clouds in models. I recommend its publication in "Atmospheric Chemistry and Physics" after the following comments are addressed.*

We are thankful to the reviewer for reviewing our manuscript, appreciating our efforts and providing us useful comments/suggestions to improve this manuscript.

Below, we have addressed the comments in a point-wise fashion. Referee comments have been given in italics and in red color. Our responses are in black.

Almost all the corresponding changes in the manuscript are marked in blue color.

**Major Comments:**

1. *Line 100: The subsection "Treatment of aerosols" can be improved. Where were the biological and non-biological aerosols derived from? Are any of these sources biomass burning? CAM6 uses MAM4 (Liu et al. 2016) for the treatment of aerosol species and processes. Was this entirely replaced by the scheme provided by Phillips et al. (2009) or only some processes? What are the seven modes in the aerosol model called? Please provide additional information.*

   We have duly improved the subsection on aerosols. The biological and non-biological aerosols are diagnosed from the insoluble organic species of MAM4, as there is no dedicated species for biological aerosols in MAM4. We simply assume half of the mass is biological and the rest non-biological.

   Generally, MAM4 is partially retained but the linkage with cloud properties is changed. We retain all aspects of MAM4 that do not pertain to aerosol-cloud interactions.

   We thank the reviewer for their suggestion. Now the section is explained in extra text (lines 126 to 164).

2. *Line 271: It looks like the precipitation between May 12th and 13th is being largely overpredicted by LS24 (Figure 1a). What processes are causing that? It also seems like the initiation of the precipitation is off by a few hours in all the simulations. Do the authors know a possible explanation for that? It would be useful to add some discussion for that in the text.*

We thank the reviewer for this important observation.

We have written an explanation of this in the text. We think the essential problem is that the stratiform precipitation following the main convective peak is happening too late in the model. The problem is one of timing. Instead of the convective outflow being a simultaneous source of precipitating stratiform cloud, as in natural MCSs (see Houze 1993, textbook "cloud dynamics"), the layer-cloud seems to build up later from the environment almost independently of the convection. Figure 1b shows that the LS24 scheme produces stratiform precipitation at the ground mostly about 12 to 15 hours too late, when observations show that all precipitation (both convective and stratiform) should fall during the major peak (11 May).

In the absence of strong production of layer-cloud by convective outflow, the humidity in the environment builds up and is removed slowly by stratiform cloud. The moisture that should be sent to the layer-cloud for conversion to stratiform precipitation instead somehow is returned to the environment during the major peak. The excess moisture in the environment then is converted to stratiform cloud almost independently of the convection. Qualitatively, a similar bias in timing, with artificial decoupling between the convective precipitation and the stratiform precipitation is seen in the original unmodified MG08 simulation.

It might also be that the mixed-phase microphysical processes of precipitation production are too weak in the stratiform cloud simulated, resulting in surface precipitation that is delayed. Sensitivity tests (not shown) indicate that disabling the bin-emulated aggregation reduces the May 12-13 peaks only slightly by 25%, suggesting this process is handled too weakly in the model. We know that most of the surface precipitation in this case is from the ice crystal process, which involves aggregation (Gupta et al. 2023).

The delay of about 6 hrs in precipitation initiation in the major convective peak, common to both schemes, stems primarily from limitations of the dynamical framework of the deep convection treatment in the global model (the ZM scheme) rather than microphysics deficiencies.

We have added text to clarify (lines 389 to 394).

3. *Figure 4: It appears that the "no-SIP" case does not really cause any marked difference in the ice-phase cloud properties relative to the control simulation (Fig. 4). Is this because the*

*model is not sensitive to SIP processes or is SIP physically not very important for ice-phase and mixedphase processes? Can the authors please provide some explanation for that?*

The relatively modest no-SIP sensitivity in this case does not indicate SIP is unimportant globally. Rather it reflects the fact that homogeneously nucleated ice is downwelled too far towards the freezing level in the simulation (c.f. Waman et al. 2022). Also, the convective outflow from cores to produce the deep stratiform layer-cloud seems too weak, as evinced by the stratiform precipitation happening long after the convective cores have decayed. These are model biases that are still to be investigated.

But also, the primary ice concentration is quite substantial (see Waman et al. 2022), leaving less potential for secondary ice to be generated, in the build-up of total ice concentrations until water saturation is attained.

New text about this has been added (lines 481 to 503).

**Minor Comments:**

1. *Line 71: The double-moment stratiform microphysics options in CAM6 are MG1 (Morrison and Gettelman, 2008) and MG2 (Gettelman et al. 2015), and not Morrison and Grabowski (2008). Please check the citation.*

   We are thankful to the reviewer for pointing this out. Now, we have corrected this (lines 60 and 88).

2. *Line 76: "were omitted Liu and Penner (2005)". Typo?* Sorry for that, the correction has been made in the manuscript (line 90).

3. *Line 76: Is the citation correct? Liu and Penner (2005) studied Primary Ice Production mechanisms via homogenous and heterogeneous nucleation modes. As far as I know, they did not discuss any mechanism for SIP. Please check.* We agree. Now the corrections have been made in the manuscript (line 90).

4. *Table 1: Why are there different sigmas for each species? Do these sigmas correspond to different aerosol modes? Also, do the geometric mean diameters correspond to the sizes at which these aerosols are being emitted?*

   Lognormal size distribution parameters for the seven aerosol species are specified in the microphysics scheme for each aerosol species. In the Table, for species with multiple modes, parameters are listed for each mode separately. The standard deviation ratio ($\sigma$) represents the width of the size distribution and varies by species based on observed aerosol properties (Phillips et al., 2009). Larger $\sigma$ indicates broader size distribution. Geometric mean diameters ($D_g$) represent characteristic particle sizes for each mode and are derived from field

campaign measurements (DeMott et al., 2003; Richardson et al., 2007). For sea salt and sulphate, $D_g$ varies with altitude above the planetary boundary layer to represent processing and aging effects.

The Table has been revised and new text has been added to explain the treatment of aerosol size distributions (lines 126 to 164).

5. *Eq. 3: Please cite Morrison et al. (2005) for the "$\lambda_x$" parameter.* We are sorry for this error. Now the citation has been added corresponding to the definition of "$\lambda_x$" (line 183).

We now realise the formula needed further explanation as for some species the bulk density varies with size, so that a lookup table is used instead (snow and graupel), (line 184).

6. *Eq. 5: How do you obtain nx,0?*

For most species, the intercept parameter, $n_{x,0}$, is related to the number concentration ($n_x$) and slope parameter ($\lambda_x$) through the zeroth moment of the gamma distribution:

$$n_x = n_{x,0} \times \Gamma(1 + p_x)/\lambda_x^{(1+p_x)} \tag{1}$$

Rearranging:

$$n_{x,0} = n_x \times \lambda_x^{(1+p_x)}/\Gamma(1 + p_x) \tag{2}$$

This relationship ensures that the discretized bin distribution (which uses $n_{x,0}$) reproduces the bulk number concentration ($n_x$) predicted by the model. In the emulated bin approach, $n_{x,0}$ is recalculated at each timestep as $n_x$ and $\lambda_x$ evolve, maintaining consistency between the bulk and bin representations.

Extra explanation is provided (line 170).

7. *Line 138: Do you mean "in-cloud droplet activation follows Abdul-Razzak and Ghan (2000)"?*

No, certainly not.

The Abdul-Razzak and Ghan (2000) scheme was for cloud-base activation of droplets. Cloud-base schemes cannot be used to treat in-cloud activation, due to differences in the supersaturation evolution. In nature, there is typically a peak in supersaturation near cloud-base (about 10 or 20 metres above it) due to a non-equilibrium overshoot beyond the quasi-equilibrium value aloft. In the interior of the cloud, the supersaturation evolves smoothly according to a quasi-equilibrium (see Eq 7.22 of Rogers and Yau 1989).

We no longer use anywhere in CAM the Abdul-Razzak and Ghan (2000) scheme.

8. *Wouldn't k-Kohler theory be used to only estimate the activation supersaturation?*

   Yes.

   It is true that kappa-Kohler theory only yields the critical supersaturation of activation of any given aerosol particle.

9. *In-cloud activation would require calculating the maximum ambient supersaturation for activation, which we cannot get simply from the k-Kohler theory.*

   Yes and no.

   Yes, the ambient supersaturation is used to predict aerosol activation in-cloud and is compared with the critical value for activation from kappa-Kohler theory.

   No, it is not the "maximum" cloud-base value of this ambient supersaturation that is used for in-cloud activation. Rather, it is the quasi-equilibrium aloft of the in-cloud ambient supersaturation that is used for in-cloud activation.

   All of this is done by our scheme described in the present paper.

10. *Also, is there a reason why Ming et al. (2006) was not used for in-cloud activation?*

    Yes.

    As noted above, one cannot apply a cloud-base scheme (whether Ming et al. or Abdul-Razzuk and Ghan) to treat in-cloud activation, because the supersaturation in-cloud follows a different evolution relative to that at cloud-base, as noted above. For clarity, one can see the vertical profiles of supersaturation given by Rogers and Yau (1989).

    In summary, only cloud-base droplet activation is computed with the scheme of Ming et al. (2006), which involves prediction of the peak supersaturation just above cloud-base arising from the non-equilibrium overshoot. One cannot apply such a cloud-base activation scheme to treat in-cloud activation as the supersaturation in-cloud aloft follows approximately the quasi-equilibrium (e.g., see Eq 7.22 of Rogers and Yau 1989 for the equilibrium in liquid-only cloud) value, which is much less than the peak cloud-base value. The in-cloud activation scheme involves kappa-Köhler theory, which is used to derive the critical supersaturation of each aerosol bin, while the ambient supersaturation on the model vertical grid is obtained from the time-dependent supersaturation solution following Korolev and Mazin (2003).

    For in-cloud droplet activation, Instead of assuming an instantaneous approach to a prescribed peak supersaturation, the supersaturation field is solved explicitly using the analytic Korolev–Mazin (2003) solution, which balances adiabatic cooling against diffusional growth onto existing liquid and ice particles. The supersaturation solution is subcycled with a timestep $\Delta t_{fine} \sim \tau_S/10$, where $\tau_S$ is the supersaturation relaxation time computed from the

Korolev solver. Within each substep, the activation scheme is applied, and activated aerosol is removed from the environmental distribution. This approach captures the temporal evolution of supersaturation within the model microphysics timestep and yields more realistic activation rates, especially under strong updrafts, while remaining numerically stable.

New text to clarify all of this treatment of in-cloud activation is added (lines 211-249).

11. *Line 255: Were the temperature and wind fields also relaxed to the observations for the SCAM6 simulations? If these are atmospheric isotherms, I suggest the plots to be made with respect to isobars or vertical height.*

The built-in SCAM nudging routine was not invoked, which would otherwise have relaxed the thermodynamic state towards observations. this is clarified with extra text (line 355).

12. *Line 263: What does "LS25" stand for?*

Sorry for the confusion, in the whole manuscript our improved/modified version of the code is referred to as "LS24". This means "Large Scale 24", our model created in 2024. Also, it is referred to as the "control simulation".

13. *Line 270: Figure 1(a) and (b) have not been separately referenced when being discussed.*
We are thankful to the reviewer for their suggestion. Now the added/modified text is given to refer to Figure 1b (lines 384 to 394).

14. *Line 271: What does "LS24" stand for?*

See the explanation above.

15. *Figure 2: What are the temperatures on the y-axis? All the text and marker sizes in the plot can be increased for better readability. Furthermore, adding standard errors/standard deviations for the simulated LWC and CDNC would also be useful.*

We are sorry for not being clear. Now the enhanced plot with the clear ranges of the standard deviations along with biased data is attached in the manuscript.

16. *Line 320: The limit for the homogenous freezing is mentioned as -36C, but In section 2.2.7, it is mentioned as -35C. Is it the former or latter? I would suspect 1C difference at these temperatures should substantially change the INPs. Can the authors please confirm?*

We appreciate the concerns raised by the reviewer. This was overlooked by us, now we will consistently mark this as "about $-36^{O}$C". In reality, the homogeneous freezing temperature depends on drop size and time, as homogeneous freezing is a stochastic process. For reasonable times of interest, the freezing happens at temperatures ranging from $-37$ to $-36$ degC for cloud-droplets, but for large raindrops it happens at almost $-35$ degC.

17. *Line 324: What is classified as the "mixed-phase region"?*

The term "mixed-phase region" refers to the range of temperatures from 0 to about −36 degC. At these temperatures, it is possible for supercooled liquid water and ice to co-exist. To explain this further we have explained this with extra text (lines 422-426).

18. *Tables 4 and 6: What is an "Unconditional average"?* We are sorry for not explaining this. It is now detailed with text added in the manuscript (caption to Table 4).

19. *Line 388: "deg". Typo.*

Sorry about that, we corrected this and replaced all other similar instances of "degC".

20. *Line 391: I am slightly confused about the discussion regarding the ice nucleation and the distribution of the ice particle number concentrations. Homogenous freezing would be "effectively turned off" above the -35C mark where the ice particles would be greater in number in the control simulations. But shouldn't the aerosol-induced freezing result in a larger ice number concentration at warmer temperatures? Can the authors please explain.*

Well, in the simulation, this does not happen and it is counter-intuitive. When the INP aerosol concentration is boosted by a factor of 100, the homogeneous freezing of solute aerosols (above the -35 degC level) is switched off, because the INPs activate in the mixed-phase stratiform ascent, causing the humidity to be depleted (e.g. by evaporating or freezing). Homogeneous aerosol freezing aloft requires humidities approaching (but not above) water saturation and hence is stopped.

Generally, solute aerosols (e.g. 10s or 100s per cm3) are orders of magnitude more prolific than IN aerosols that are rare (e.g. perhaps 10 per Litre). Ordinarily, in the control run, even if only a small fraction of the high numbers of ice crystals from homogeneous aerosol freezing aloft are downwelled into the lower half of the mixed-phase region, this is enough to outnumber the primary ice from heterogeneous ice nucleation. Homogeneously nucleated ice seems to prevail throughout the mixed-phase stratiform cloud in the control simulation.

In the high (100 x) INP run, this downwelling of homogeneously nucleated ice is switched off. Hence, there is the reduction of ice concentrations throughout the mixed-phase region relative to the control run. We think there may be a model bias with too much downwelling of homogeneous ice compared to our high-resolution cloud model simulations and not enough convective outflow into the stratiform cloud.

We have added fresh text to clarify about this (line 547).

21. *Line 444: I am not sure which scheme is LS25, the discussion in the manuscript only mentions LS24 throughout the writeup*

Sorry this was been overlooked by us, now corrected throughout the manuscript.

**References**